# Interactive Deep Learning for Shelf Life Prediction of Muskmelons Based on an Active Learning Approach

**DOI:** 10.3390/s22020414

**Published:** 2022-01-06

**Authors:** Dominique Albert-Weiss, Ahmad Osman

**Affiliations:** 1University of Applied Science htw saar, 66123 Saarbruecken, Germany; d.albert-weiss@posteo.de; 2Fraunhofer Institute for Nondestructive Testing IZFP, 66123 Saarbruecken, Germany

**Keywords:** deep learning, quality monitoring, agriculture, active learning

## Abstract

A pivotal topic in agriculture and food monitoring is the assessment of the quality and ripeness of agricultural products by using non-destructive testing techniques. Acoustic testing offers a rapid in situ analysis of the state of the agricultural good, obtaining global information of its interior. While deep learning (DL) methods have outperformed state-of-the-art benchmarks in various applications, the reason for lacking adaptation of DL algorithms such as convolutional neural networks (CNNs) can be traced back to its high data inefficiency and the absence of annotated data. Active learning is a framework that has been heavily used in machine learning when the labelled instances are scarce or cumbersome to obtain. This is specifically of interest when the DL algorithm is highly uncertain about the label of an instance. By allowing the human-in-the-loop for guidance, a continuous improvement of the DL algorithm based on a sample efficient manner can be obtained. This paper seeks to study the applicability of active learning when grading ‘Galia’ muskmelons based on its shelf life. We propose *k*-Determinantal Point Processes (*k*-DPP), which is a purely diversity-based method that allows to take influence on the exploration within the feature space based on the chosen subset *k*. While getting coequal results to uncertainty-based approaches when *k* is large, we simultaneously obtain a better exploration of the data distribution. While the implementation based on eigendecomposition takes up a runtime of $O(n^{3})$, this can further be reduced to O(n·poly(k)) based on rejection sampling. We suggest the use of diversity-based acquisition when only a few labelled samples are available, allowing for better exploration while counteracting the disadvantage of missing the training objective in uncertainty-based methods following a greedy fashion.

## 1. Introduction

With increasing need for sustainability within the food supply chain, the quality control and food monitoring of agricultural commodities based non-destructive testing has gained in valuable interest [1,2,3,4]. This particularly relates to ’Galia’ muskmelons where its climacteric nature results in the contamination of neighbouring fruits when one is afflicted with diseases or pesticides [5]. While the research for agricultural goods has focused on computer vision [6,7,8,9,10], recent works have used other non-destructive techniques such as X-ray [11,12], biochemical sensors [13,14,15] and IR spectroscopy [16,17,18,19], giving information of the interior. These techniques are either very time-consuming or can only be achieved in a costly and labour-consuming manner. In contrast, we propose the use of acoustic resonance testing (ART) allowing for a rapid, global and cost-effective state evaluation of the agricultural goods.

ART is a well-established method for testing the resonance properties of agricultural commodities, profiting from the mentioned benefits without having to do extensive sample preparation or laborious calibration. While it has been used as a reference for determining the material stiffness by studying the natural resonance frequencies based on vibrational properties [20], studies have shown that the acoustic properties also relate to the water content [21,22,23] and the crispiness [24,25] of the product using a texture analyzer [26] or conducting assessments based on consumer panels [26]. While crispiness is a haptic feature associated with the firmness and the degree of freshness, these provide high correlations with the auditory perception. Based on a physical perspective, these acoustic deviations derive from shifts of the signal response. Thus, with the change of the acoustic signal response, the resonance frequencies can be associated with the acoustic fingerprint of the fruit ripening.

While the choice of algorithm is crucial in deciding the generalisation capabilities, former studies in ART have used an experimental approach to validate the ripening of agricultural products. This is highly impractical since biological materials are inhomogeneous in nature and come with different shapes and sizes. Deep learning (DL) serves these needs allowing for a high adaptability to novel data based on the universal approximation theory [27,28,29]. Specifically the use of CNNs have emerged as a well established technique for imagery data, having lead to the use of a wide variety of computer vision tasks. Characteristic for CNNs is the extraction of features from raw data input with specific patterns, without having to apply manual feature design. Within the recent years, DL has found its ways into agricultural studies for which fruit detection, ripeness classification and detection of diseases has been of great interest [30,31,32]. Sa et al. [33] developed a fruit grading system by evaluating the fruit ripeness by surface defects using hyperspectral imaging. Zeng et al. provides a thermal imaging system for pears, studying the development of bruises with ageing [34]. Zhu et al. uses capsules as network design for grading carrots based on their appearance [35]. While they provided the transferability of DL to agricultural settings, the accessibility of sufficient amounts of data was assumed as a given prerequisite.

Active learning (AL) is well-motivated in a wide field of real-world scenarios where labelled data are scarce or costly to acquire. AL is a framework in machine learning allowing reducing the effort for data acquisition by making the assumption that the performance of the machine learning algorithm can be improved, when the algorithm is allowed to choose the data, which it wants to learn from [36,37]. For supervised learning, this can be extended by the idea of being provided only by a subsample of labelled data instances. The learner sends a quest to the annotator who is supposed to provide a label for an unlabelled data instance. This has traditionally been achieved by sampling instances close to the decision boundary on which the algorithm is most uncertain about, known as uncertainty sampling. Afterward, the model is retrained based on the augmented training instances [38]. This procedure is repeated until the labelling budget is exhausted or a predefined stopping criterion is met [39].

While AL has been of great interest in machine learning, the application to CNNs has only begun within the recent years [37,40]. The reasons for this trace back to the question on how uncertainty should be modelled [37]. This question has been encountered by using the softmax response to represent the activation of an instance, opening new questions in regard of the effectiveness of the emerged heuristics applied in a batch aware setting. Uncertainty-based methods have been favoured with the argument that diversity-based methods might be prone to sampling instances in sampling space that do not lead to additional information gain [41]. Furthermore, diversity-based sampling strategies are accompanied with a high computational expenditure. We propose k-DPP as a purely diversity-based method with the argument that computational complexity is becoming less of an issue for the 21st century, where computational resources become readily available. We are able to show that no substantial information is lost which can be observed during the training of AL, resulting in equally good results.

The main contribution of this paper is to show that AL can significantly enhance DL algorithms in agriculture setting by reducing sample effort or artificially generating an overfitting when labels are weak. This is done by comparing different acquisition functions to random sampling as a baseline, which has often been challenging to surpass in the context of AL applied to DL architectures [42,43]. We introduce *k*-DPP as a novel acquisition function and show that it is a suitable sampling technique in AL, obtaining on average competitive results for accuracy, recall and precision. This is especially true for large *k* allowing for a better capture of the data distribution by considering a larger subsample during the sampling process. This allows to surpass sampling bias often introduced by uncertainty-based methods [41,44,45], while coming with the advantage of easily being parallised.

In summary, the paper delivers the following novel ideas:guidance and sample efficient improvement of the DL algorithm through a human-in-the-loop setting,potential reduction of the required samples for training DL algorithm andintroduction of *k*-DPP as a diverse sampling method with improved capture of the underlying data distribution compared to uncertainty-related metrics.

To our knowledge, this is the first time deep AL is applied for monitoring quality and ripeness in agriculture.

## 2. Materials and Methods

### 2.1. Data Collection

For the experiments, 30 fresh ‘Galia’ muskmelons (*Cucumis melo*) of different sizes and shapes were shipped to Germany after being harvested from different farm regions. Samples originated from both Spain and South America, allowing the consideration of diversity in growing conditions. ‘Galia’ muskmelons were provided during a time span of two months including March and May; the initial weight at arrival ranged from 837.2 g to 1555.3 g. The weight was measured via a high-precision scale.

During the shipping, the fruits were cooled at a temperature range of 9–12 °C, which is considered a typical cooling condition for handling melon fruits [26]. The Galia muskmelons arrived at a immature stadium. To study the ripening, the galia melons were stored at room temperature ranging between 18.4 and 22.9 °C. While being held in batches, it was assured that during the measurement period, the ‘Galia’ melons were stored in the same batch for better tracing of cross-sensitivity of defects and pesticides. For better distinction between the different samples, the ‘Galia’ fruits were provided with a tag for unique identification, allowing for traceability during the measurement periods.

A panel was executed for tasting the fruits so that a better distinction and judgement of the different classes could be made. This should ensure better discrimination of the classes and, thus, should give a better understanding about the output of the DL algorithm. For the selection of the fruit piece, the melon was cut in the middle where both an outer and inner piece was tested. To minimise the variance regarding how the taste of the fruit piece is perceived, only one person was selected for the panel, maintaining the same diet through these months.

### 2.2. Experimental Design

Situations in which labelled data is cumbersome to obtain, active learning has proven a valuable technique to acquire labelled samples with the potential to reduce the amount of data required for training deep learning algorithms. While uncertainty-based methods have been extensively studied especially in traditional machine learning, we argue that the training objective can easily be missed especially in deep learning settings while often leading to sampling bias. For this reason, we propose *k*-DPP as a diversity-based methods. Opposed to traditionally used acquisition functions, *k*-DPP samples throughout the feature space allowing for a better capture of the data distribution. We anticipate better results in comparison to random sampling, as clustering of sample points is obviated by a negative correlation between sample points and, thus, is especially advisable when the number of collected samples is small.

To support this assumption in agricultural settings, measurements were executed by studying the resonance properties using acoustic testing. During the time span of five weeks, the acoustic sound profile was recorded once a week by a single-channel microphone. The specimen is brought in vibration by a hammer made of silicon nitride, for which a single-channel omnidirectional condensator microphone was placed directly at the excitation position. The sampling rate was at 44.1 kHz. The muskmelons were knocked at six different positions, considering four positions in the middle as well as the two ends of the fruit. Measurements were repeated five times per location. As the material density directly impacts the resonance frequencies, the fruit’s weight was recorded. In addition to that atmospheric parameters such as the room temperature and humidity were measured for handling the sensitivity of the microphones.

The application of active learning presumes that a neural network architecture is trained in advance. Thus, based on the collected data, a convolutional neural network was used for this purpose. To improve training results, preprocessing of the collected data was obtained by applying a gain filter for speech enhancement. Furthermore, before feeding the data to the neural network, the signals where shifted to the same starting point, cropped and converted in frequency domain. This is discussed in more detail within Section 2.4. Unlike traditional processing, we applied a convolutional neural network which has become more prominent for evaluating acoustic data within the recent years [46].

As a primary building block, the neural network is retrained during the active learning cycle. To capture the relevant instances for querying, the choice of acquisition function is of predominant choice and should be addressed in depth within this paper. While proposing *k*-DPP as a diverse sampling technique, we explore different acquisition functions using the same network architecture. Although diversity-based methods might come with the expense of additional computational effort, we justify the choice of sampling technique by demonstrating that we obtain equivalent to superior results when *k* is large.

### 2.3. Related Fields of Research

While traditionally deep learning requires enormous amount of data, recent interest in deep learning traces back to evaluation methods allowing a reduction of labelled instances. For clarity, we want to differentiate from other related fields such as imitation learning, zero- and few-shot learning as well as self-supervised learning.

Original applied in robotics, imitation learning tries to imitate the behaviour of a labelled dataset collected by an expert while trying to learn a policy that imitates this behaviour based on some set of trajectories [47,48]. For instance, behavioural cloning performs a supervised learning task by learning from observations, while inverse reinforcement learning optimises for an ideal reward function [49]. Active learning and imitation learning relate to each other in the way that both preassume that the labelling is achieved by a human being with expert knowledge. However, while imitation learning assumes that the algorithm adapts the behaviour of expert, active learning queries novel instances to reach further information gain [50].

A novel approach for classifying instances based when few or even none instances are available is Few-shot and Zero-shot learning [51]. Few-shot learning is another subfield in deep learning used within the low data regime, making the assumption that classes are disjoint [52]. Here, the relation between classes is usually learned based on the similarity in the embedding space. Instead of learning the label itself, the latent relationship between instances is learned. However, while zero- and few-shot learning optimise for instance the similarity between classes in the latent space, active learning allows optimising for the relation between input data and the label itself.

Another arising field within the last years is self-supervised learning. Just as in active learning, self-supervised learning is given both a labelled and unlabelled dataset for which the percentage of labelled instances very small. It is largely motivated by the idea of learning meaningful representations by optimising the unlabelled dataset based on on the labelled one [53]. In other words, a unsupervised learning task is solved under supervision [54].

### 2.4. Dataset

The dataset follows the data collection and experimental design of Section 2.1 and Section 2.2, while the acoustic profile for each fruit was measured once a week. Regarding the shelf life *s*, measurements where executed at s∈{0,7,10,15,17,62} days after arrival. In addition to that, the dataset consists of meta information such as the weight, location of knocking, room temperature and humidity which serve as auxiliary variables to the algorithm.

For better convergence during training of the neural network, a Single-Channel Noise Reduction (SCNR) algorithm was applied, visualised within Figure 1. Designed for the speech enhancement of single-channel systems, a noise reduction is achieved by computing the gain filter based on the calculation of the signal energy and noise floor during every frequency bin. Assuming a signal with *n* samples, the gain *G* is calculated for the *k*-th frequency bin by [55,56]
(1)$$G\left[k,n\right]=max\left\{{\left(\frac{P\left[k,n\right]-\beta P_{N}\left[k,n\right]}{P[k,n]}\right)}^{\alpha },G_{min}\right\},$$
where $G_{min}=10^{-DB_{red}/20}$ is the minimal gain and P[k,n] represents the energy calculated by squaring the frequency amplitude. $DB_{red}$ is the maximal noise reduction to be performed per bin to avoid the occurrence of *musical noise*, set at $DB_{red}=25$ dB as default. To calculate the energy of the noise estimate, $P_{N}$ can be calculated by determining the lowest energy based on the window size *L*:(2)$$P_{N}\left[k,n\right]=\min_{\left[n-L,n\right]}P\left[k,n\right]$$

Furthermore, α=9 and β=45 were selected for suppression of artefacts. As seen in Figure 1, while this shows to be an effective method for reducing broadband noise, artefacts are added by including new frequencies at certain time frames which were previously unseen. However, as the signal is cropped to the region of interest, this side effect had little influence on training process of the DL model. Before feeding the data to the network, normalisation of the auxiliary variables is applied while the amplitude *A* and the phase ϕ have been standardised.

Before feeding the data to the AL setup, the data were split into training, test and validation sets, visualised in Table 1. The dataset distinguishes between four different classes: y∈{1,2,3,4}. These are differentiated based on the shelf life of the fruit for which each class deviate by one week in terms of the storage time. Data augmentation based on vertical and horizontal flipping of the raw acoustic signal was applied, resulting in 8080 data instances. Besides the acoustic signals, weight, room temperature, humidity, as well as the location of knocking were enclosed as auxiliary variables.

In reference to the executed panel discussed within Section 2.1, it showed that class 1 consists of ripe and eatable fruits with the need of further aroma development. Class 2 and 3 seemed eatable, while class 4 contained eatable portions consisting of at least one overripe region. Images of the ‘Galia’ muskmelon for the different classes can be found in the Section A.1.

### 2.5. Active Learning Framework

Acquiring labelled data turns out to be costly especially when DL is applied. AL is a framework allowing for a reduced data acquisition when samples are scarce. This has traditionally been approached by sampling based on instances it is most uncertain about, actively querying for unlabelled data instances [36].

Figure 2 demonstrates the iterative process of the AL framework in a DL learning setting. Based on the unlabelled data instances, the learning algorithm may pose queries to an oracle which is usually a human operating as an annotator. This is usually done by choosing an uncertainty-based sampling strategy, where data instances which are located close to the decision boundary, are more likely being queried. For a pool-based scenario, let us assume a set of data instances $X=(x_{1},\ldots ,x_{N})$ for which $x_{i}$ represents the data instance itself. Further, let us assume $Y=(y_{1},\ldots ,y_{N})$ being the corresponding set of labels, for which $y_{i}$ represents the label for *i*-th data instance. The neural network is retrained based on the labelled training set L augmented each iteration by the pool set U which has previously been annotated by an oracle [36,38]. This procedure is repeated until the labelling budget is exhausted or a predefined stopping criterion is reached [39,57]. It is to note that AL is traditionally formulated by sampling on one instance per iteration. While training on a single instances at a time is ineffective in a DL setting and may also favour overfitting, sampling in batches has been preferred. Thus, the sampling is executed in batches such that $B=(x_{1},x_{2},...,x_{b})$ for which b<N[41]. When speaking of a *batch-aware* setting for DL, the parametrised classifier $h_{\theta }$ can be stated by
(3)$$h_{\theta }=\underset{\theta }{\mathrm{argmax}}f{\left(B;\theta \right)}_{y}\;\mathrm{for}\;B\subseteq L.$$

### 2.6. Choice of Acquisition Function

The design of an adequate acquisition function is decisive in terms of the minimisation of the labelling cost, playing a crucial role in the success of the AL framework. However, for deep neural networks a clear understanding of uncertainty is not straightforward. Methods used in the classical sense such as optimal experiment design [58] would be intractable for CNNs as it requires the computation of the inverse of the Hessian matrix during each training iteration [42]. Another issue is the increasing time complexity required of some algorithms with increasing dimensionality [42]. Thus, to resemble uncertainty in the sense of neural networks, the softmax response is used to approximate the activation of an instance [59].

While uncertainty-based acquisition functions has been mostly investigated, recent studies have also studied the impact of exploratory approaches based on diversity-based sampling. To reduce the cost of annotation the selection of the most representative and informative samples is required. Within our study, six different acquisition functions were assessed consisting of random sampling as the baseline, four uncertainty-based sampling methods and *k*-DPP as a diversity-based approach. Although there has not been a clear definition regarding an informative, representative sampling, informativeness has been associated with uncertainty while representativeness concentrates on diversity. Diversity-based methods select data points diversely throughout the feature space, compensating the lack of exploration within uncertainty-based methods. This has usually been tackled by geometric approaches, such as coresets [37] or compression scheme that uses farthest-first traversals [59]. The authors of [60] followed a hybrid approach by incorporating uncertainty and diversity by sampling points represented in a hallucinated gradient space without tuning an additional hyperparameter. With the introduction *k*-DPP we provide a novel approach that is not only easy to parallelise but also offer possibilities to reduce computational complexity.

Within this study the following acquisition functions should be investigated:**Random acquisition** represents the baseline where instances are sampled stochastically without the heuristic calculation of a metric.**Least Confidence** samples the instances where the algorithm is least confident about the label and is calculated by
(4)$$X_{LC}^{U}=\underset{x}{\mathrm{argmax}}\left(1-P_{\theta }\left(\hat{y}|x\right)\right)$$
with posterior probability $P_{\theta }\left(\hat{y}|x\right)$.**Margin sampling** calculates the margin between the most probable and second most probable classes represented by $\hat{y_{1}}$ and $\hat{y_{2}}$:
(5)$$X_{MS}^{U}=\underset{x}{\mathrm{argmin}}\left(P_{\theta }\left({\hat{y}}_{1}|x\right)-P_{\theta }\left({\hat{y}}_{2}|x\right)\right)$$**Maximum entropy** samples instances yielding the maximum entropy by determining
(6)$$X_{E}^{U}=\underset{x}{\mathrm{argmax}}\sum_{i}P_{\theta }\left(y_{i}|x\right)\cdot log(P_{\theta }\left(y_{i}|x\right).$$**Ratio of confidence sampling** is very closely related to margin sampling where the two scores with the highest probable classes are determined as a ratio instead of the difference.**Bayesian Active Learning of Disagreement (BALD)**: The goal of BALD is to maximise the mutual information I between the prediction and model posterior such that, under the prerequisite of being Bayesian, BALD can be stated as
(7)$$I\left(y;\theta |x,L\right)=H\left(y|x,L\right)-E_{p(\theta |L)}\left(y|x,\theta ,L\right)\left[H\left(y|x,\theta ,L\right)\right]$$
with H representing the entropy. Instances leading to a maximal score relate to points on which the algorithm is uncertain on average and for with the model parameters are referenced with a high disagreeing prediction [61,62].**k-Determinental Point Processes (k-DPP)**: As an diversity-based approach, *k*-DPP takes an exploratory approach by sampling based on the DPP conditioned on the modelled set being of cardinality *k* [63] (We like to note that we prefer the use of *k*-DPP over traditional DPP due to the introduced bias into the modelling of the content. Parameter *k* allows to take a direct influence on the diversity by taking into regard the repulsiveness of the drawn samples—or in other words—the magnitude of the negative correlation between samples).

To define a point process P, let us assume a random set Z being drawn from P for which A⊆Z and Z being a discrete set. Given some positive semi-definite kernel matrix $K_{A}$, the sampling process within *k*-DPP can be described by sampling *k* out of *n* subitems for which the probability is proportional to the determinant of the kernel spanned over indices contained within the elements of *A* [63,64]. Thus, we can write
(8)$$P\left(A\subseteq Z\right)=det\left(K_{A}\right)$$
where *K* is some semi-definite matrix and $K_{A}$ confines *K* to the entries of the indices of *A*. To sample from *k*-DPP, given the eigenvector-value pairs $\left(v_{n},\lambda_{n}\right)$ and *k* itself, *k*-DPP can be described by a mixture components of DPP. Let J be a random variable, the marginal probability of index *N* is derived by
(9)$$Pr\left(N\in J\right)=\frac{\lambda_{N}}{e_{k}^{N}}\;\sum_{\begin{array}{c} J^{\prime }\subseteq \left\{1,...,N-1\right\}\\ |J^{\prime }|=k-1 \end{array}}\;\prod_{n\in J^{\prime }}\lambda_{n}=\lambda_{N}\frac{e_{k-1}^{N-1}}{e_{k}^{N}}$$

For more information, we refer to [63].

## 3. Results

### 3.1. Model Training

For more robust and meaningful results in regard of the findings obtained via AL, a CNN was trained, visualised within the architecture shown in Figure 3. Before feeding the amplitude *A* and the phase ϕ to the CNN, it was ensured that the data were preprocessed by the SCNR algorithm and the signals were aligned based on the maximum argument. To avoid feeding redundant information to the neural network, a cropping to the region of interest was applied. Additionally, the processed signals and the auxiliary variables were shuffled for better training performance. After concatenation of the amplitude and phase spectrum, the information is given to the next layers, consisting of four modules comprising a batch normalisation layer, two convolutional layers, one max pooling layer and one dropout layer with probability of 0.005. Batch Normalisation was used in both architectures, allowing to normalise activations of the batch within the intermediate layers [65] which has been brought in correlation with a smoother learning landscape [66]. Regularisation by using dropout has been used in favour of dropping connections to the succeeding layer [67]. Stochastic gradient descent (SGD) showed to be a valid optimiser when using momentum *m* of m=0.9 and being nestorov activated [68,69]. After compression is achieved by the convolutional layers, flattening was applied, allowing for concatenation with the scalar auxiliary variables weight, temperature, humidity and the location of knocking, referred to as zone in Figure 3. For both models a learning rate of 0.005 was effective, run with a learning rate scheduler showing exponential decay for faster convergence. Due to exploding gradients, additional gradient clipping was used at 1.0 as well as rescaling of the norm of the gradient. To handle the problem of exploding gradient, the activation function *LeakyRelu* was used in replace of Rectified Linear Unit (*ReLU*) throughout the model [70]. An illustration of the architecture can be found within Figure 3. For further interest, we refer to Section A.2, summarising the preprocessing and hyperparameters for clarity.

### 3.2. Training Setting for Active Learning

Purpose of this study is to examine the operability of AL to acoustic data recorded in laboratory environments for a pool-based scenario [36]. Centrepiece of the AL framework is the choice of the acquisition function. While uncertainty-based methods has often been used in favour for reducing the amount of labelling time, we introduced *k*-DPP as a diversity-based approach, which should be thoroughly studied in Section 2.3 by analysing different values for *k*.

For training the AL cycle, we start with an initial labelled dataset $L_{0}=30$ of randomly selected instances. In addition to that a pooling set of U=50 was selected, representing the set of data being labelled during each training loop. To study the informativeness and the representativeness of the pooled data, both uncertainty and diversity-based acquisition functions was studied. Further, we like to thoroughly study the impact of *k* in *k*-DPP which should be juxtaposed to the uncertainty-based acquisition functions. To validate the robustness of the sampling method, the training was executed over five trials.

### 3.3. Diverse *k*-DPP Sampling

To evaluate the deployability of *k*-DPP as an acquisition function in AL, we evaluate the sampling method by running multiple trials for different values of *k*, assuming the architecture from Figure 3 as the used DL model. As seen in Figure 4, a better performance is achieved when *k* is increasing, implying that a large subset per iteration leads to superior performance. This becomes more evident when comparing k=1, as it not only trains poorly for all considered metrics, but also suffers from increasing error bounds. The explanation derive from the data distribution being captured. With increasing subset *k*, the captured data distribution stronger resembles the true unknown data distribution of the dataset allowing for a better representation of the data. Thus, a higher value for *k* not only leads to better results but also higher stability.

While it is difficult to surpass the results of random acquisition for few training iterations, we specially can show for k=200 that on average a better performance is achieved throughout the course. Except for the value of k=1, the metrics converge to the same result for AL with running enough training iterations. This specifically can be seen when analysing the training curves for the accuracy and the precision at Figure 4a,d.

Noticeable after a few training iterations for both the accuracy and the recall in Figure 4a,c is that a steady increase with a sudden drop is observed. We explain this phenomenon by the idea that with the presence of only a minor sample of training data, the wrong training objective is learned. The subsequent increase is explained by an adjustment of the neural architecture to the novel data. To validate this statement we refer to future works.

### 3.4. Human-in-the-Loop Performance

When deciding for the acquisition function to be used in AL, one has to take the budget for labelling into consideration. This is specifically to mention when comparing uncertainty and diversity-based methods for querying. One reason for the establishment of more uncertainty-based query strategies can be related to it being less computational expensive while arguments were made that diversity-based methods may lead to sampling instances that result in no additional information gain [41].

Analogous to the previous section the training results for AL are evaluated based on the accuracy, loss, precision and recall during test time. To compare against *k*-DPP, the best outcome from Section 2.3. based on k=200 has been used.

Similarly to Figure 4b, we obtain good convergence for all metrics in Figure 5. As it can be seen based on the results for Figure 4a, margin sampling receives the highest accuracy, for which equal results are obtained already after half of the training iterations. Both for Figure 4a,c, a fast rise can be observed followed by a subsequent decline for both accuracy and recall. This has been explained by the few training samples available at the beginning of training which might lead to optimising for the wrong training objective. This can especially be observed for BALD in Figure 5a,c, showing stronger deviations with test time accompanied with strong error bounds.

While the optimising for accuracy may require less labelled samples, the precision in Figure 5d shows that there is potential for more data. This is different to Figure 5c where after a downshift there seems to be little improvement.

For further analysis of the results, we refer to the Table 2, summarising the results for all acquisition functions based on the average and standard deviation obtained at the last iteration. Overall, it can be said that the studied acquisition functions only surpass the baseline of random sampling by a minor extend. The best results are obtained for margin sampling outperforming the other acquisition function in accuracy and precision. Despite the similarity of the ratio of confidence to margin sampling, a large deviation in regard of the performance can be observed throughout the course of all metrics. On average, *k*-DPP reaches competitive results in comparison to uncertainty-based methods. Most noticeable is also the lower variances obtained throughout observed for both loss and precision. This can largely be explained by the large subsample *k* giving a better proxy for the true data distribution.

While margin sampling generally shows better performance in comparison to its counterparts, least confidence and *k*-DPP predominate when the stability of the metric is of interest. When addressing the stability of the acquisition functions, *k*-DPP surpasses margin sampling by showing a lower error bounds in accuracy, loss and precision.

## 4. Discussion

For supervised learning applications usually a large number of labelled data instances is assumed readily available. AL provides the possibility to actively acquire new labels from samples which comes with the expense of a human annotator being available. However, as quality inspection of agricultural goods still remains on manual inspection, AL provides a suitable machine learning framework. Taking reference to attained research goals in Section 1, we introduced *k*-DPP as a diverse sampling method coming with the expense of an additional hyperparameter in comparison to conventional DPP. Characteristic for DPP is both the modelling of the content as well as the size of the dataset. We chose *k*-DPP with the expense of another hyperparameter by getting the freedom of taking control over the content. While DPP can result in disadvantage in leading to a bias in the model of the content, *k*-DPP conditions on DPP by adjusting the model based on a subset *k* [63].

Taking reference to the mentioned research goals, we successfully introduced *k*-DPP as a competitive acquisition function. Although the results might be sensitive to the chosen architecture, it was shown that *k*-DPP delivers coequal results in comparison to uncertainty-based sampling approaches when *k* is large. Here, it is worth discussing whether uncertainty sampling is in preferred over *k*-DPP, allowing to sample diversely in space. The negative correlation between data points lead to a repulsion and thus, avoids the formation of clusters. This benefit of *k*-DPP results in a better capture of data distribution in comparison to uniform sampling. *k*-DPP not only achieves highest performance in comparison to random acquisition but also provides a high stability in comparison to the presented acquisition functions [63].

One reason for the preference of AL is the need for less labelled samples. It was shown that a reduced amount of samples are sufficient for both accuracy and recall. In the case of margin sampling, only half of the queries are needed to achieve the same performance in accuracy during test time. Nevertheless, the reduction of labelled samples was only applicable recall during testing, resulting in the advice for studying additional datasets and architectures. While it has shown that the architecture of neural networks can strongly influence the results of the acquisition function, we will consider neural architectural search within the future studies [42].

Furthermore, while *k*-DPP reaches competitive results in comparison to their uncertainty-based companions, one of its major downsides remain the increasing computational complexity. The first approaches for implementing *k*-DPP are based on eigendecomposition, which does not scale well in terms of runtime being of $O(n^{3})$[63]. While there has been newer approaches by considering the replacement of eigendecomposition by Cholesky factorisation, these allow for better numerical stability, however, without any improvement in the runtime [71]. Further approaches tried to approximate the eigendecomposition, however, with the cost of performance degradation. Ref. [64] was able to reduce the runtime to O(n·poly(k)) by using rejection sampling, generating a uniform dataset from which a proxy subset is generated following the same distribution as the true unknown data distribution. While the work of [64] is a promising technique for allowing same performance less computational complexity, their works should be considered for application opening the deployment on low-power devices.

## 5. Conclusions

The high demand in profitable crop throughput emphasises the importance of digitalised monitoring systems for the entire supply chain of agricultural commodities. While artificial intelligence has obtained high interest due to its exceptional generalisation capabilities, its data-hungry nature leads to the search for novel sample efficient methods. AL allows to dynamically query for new labelled instances, allowing deep learning to adapt to a low-data regime. In terms of the accuracy, we are able to show that all acquisition function except of BALD require half the data to obtain the same performance. Similar results can be achieved by the evaluation of the recall for which a saturation is achieved after a high peak during test time.

While the use of uncertainty-based methods may result in additional sampling bias, we proposed *k*-DPP which allows for a better capture of the data distribution and more stable results. Based on the presented works, we are able to show the potential of AL when little data can be acquired. Whereas our proposed technique can easily be parallelised, we show that it outperforms existing acquisition functions while guaranteeing stability. As uncertainty-based methods follow a greedy approach, these are more prone to missing the training objective. *k*-DPP is less susceptible to this problem by incorporating diversity based on the negative repulsion of the data instance within the feature space and, thus, enable a better approximation of the true data distribution. We are able to show that an improved representativeness can be achieved by increasing *k*, allowing for better approximation of the true data distribution and while leveraging for the exploration based on the adjustment of *k*.

The use of *k*-DPP technique comes with the expense of further computational expenditure. However, it can be argued that this becomes less of an issue as computational power becomes more readily available, and we proposed a method that allows to reduce the runtime from $O(n^{3})$ with traditional eigendecomposition to O(n·poly(k)). Furthermore, we propose a technique which allows to reduce the computation expenditure by replacing the implementation of the eigendecomposition by rejection sampling approach.

## Figures and Tables

**Figure 1 sensors-22-00414-f001:**
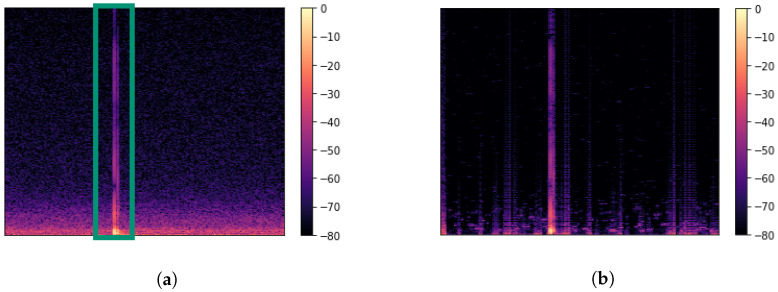
Spectrogram of a sample (**a**) before processing and (**b**) after processing based on spectral subtraction by using a SCNR algorithm with parameters β=45 and α=9. As shown in panel (**b**), a reduction of the low frequency noise is achieved without degrading the signal of interest. A side effect is the generation of artefacts by masking new frequencies at different time periods. As the signal of interest is of short period, the artefacts have little impact. The green box in panel (**a**) resembles the crop towards the signal of interest.

**Figure 2 sensors-22-00414-f002:**
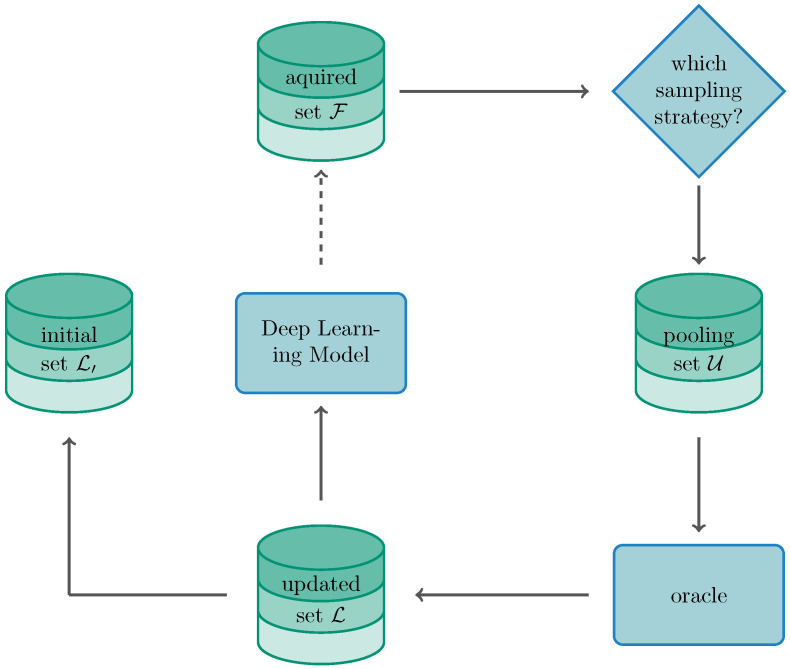
Visualisation of the active learning cycle for a DL setting.

**Figure 3 sensors-22-00414-f003:**
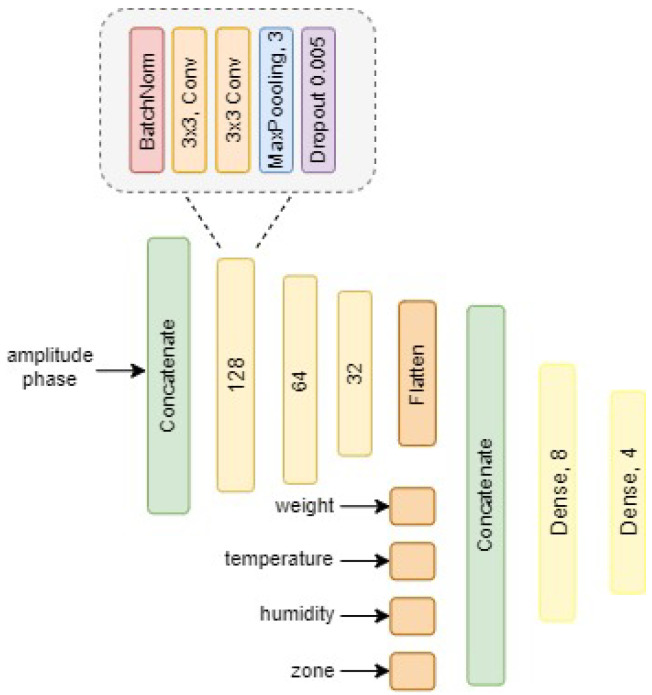
Visualisation of the DL architecture. The information extraction of the amplitude and phase results based on four convolutional modules with descending filter size towards the output.

**Figure 4 sensors-22-00414-f004:**
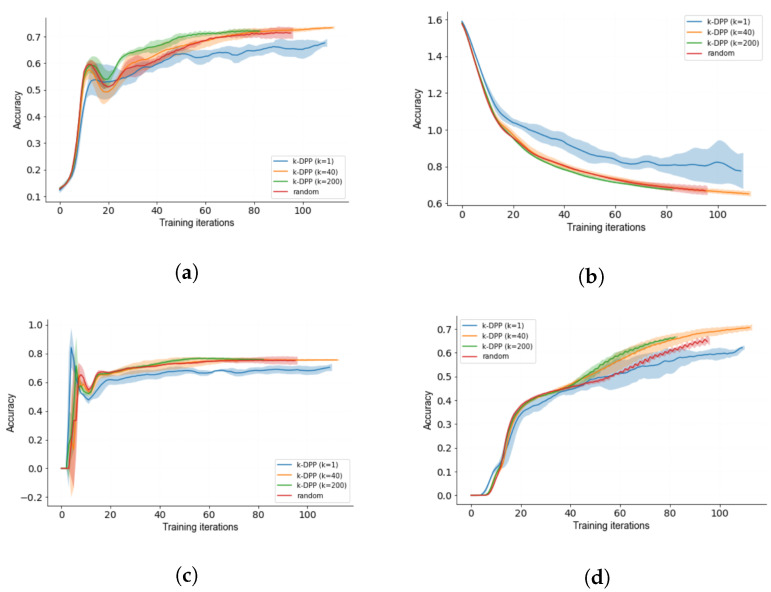
Active learning curves with error bounds for (**a**) accuracy, (**b**) loss, (**c**) recall and (**d**) precision.

**Figure 5 sensors-22-00414-f005:**
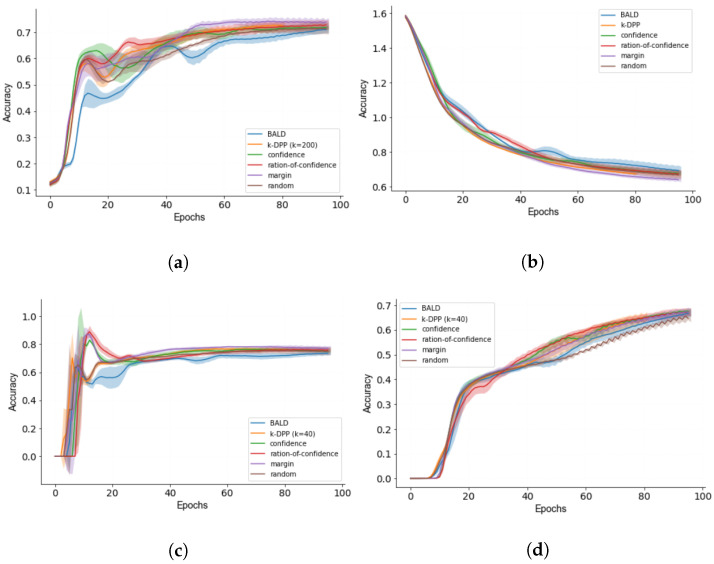
Active learning for *k*-DPP at values k∈{1,40,200,400}. Results are averaged over five iterations shown with error bounds for (**a**) accuracy, (**b**) loss, (**c**) precision and (**d**) recall.

**Table 1 sensors-22-00414-t001:** Overview of the number of data within the dataset split in into training, test and validation sets. Based on the summation of all classes $X_{train}$ consists of 0.6, $X_{test}$ of 0.25 and $X_{val}$ of 0.15 based of the total amount of data.

Class	$X_{train}$	$X_{test}$	$X_{val}$
1	968	429	259
2	1086	422	228
3	582	231	163
4	1050	454	272

**Table 2 sensors-22-00414-t002:** Summary of the studied acquisition functions in comparison to the different metrics. Within the table we represent the average and the standard deviation for the results obtained after the last iteration. The best performing acquisition metric is highlighted in by bold caption.

Acquisition Function	Accuracy	Loss	Precision	Recall
BALD	0.7098	0.6935	0.7361	0.6667
	(0.1290)	(0.0228)	(0.0132)	(0.0131)
least confidence	0.7174	0.6760	0.7531	**0.6760**
	(0.0083)	(0.0132)	(0.0116)	**(0.0103)**
*k*-DPP	0.7260	**0.6747**	0.7615	0.6504
	(0.0107)	**(0.0051)**	(0.0093)	(0.0221)
margin sampling	**0.7391**	0.7391	**0.7742**	0.6712
	**(0.0150)**	**(0.0139)**	(**0.0156**)	(0.014)
ratio of confidence	0.7283	0.7283	0.7596	0.6714
	(0.1827)	(0.209)	(0.0164)	(0.0161)
random	0.7135	0.7135	0.7509	0.6469
	(0.0220)	(0.0247)	(0.0277)	(0.0162)

## Data Availability

The dataset used in this research work can be provided upon the requests through the emails.

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
