# Peer review of "Interactive Deep Learning for Shelf Life Prediction of Muskmelons Based on an Active Learning Approach"

_sensors, 2022, doi:10.3390/s22020414_

Round 1
Reviewer 1 Report
In this paper, the authors have proposed guidance and sample efficient improvement of the DL algorithm through a human-in-the-loop setting. It produces potential reduction of the required samples for training DL algorithms. It introduces k-DPP as a diverse sampling method with improved capture of the underlying data distribution compared to uncertainty-related metrics. In general, this paper is well written and easy to follow. I would like to accept this paper if my following concerns are carefully addressed.
(1) The authors need to emphasise their contributions/novelties in the revision. In the current version, the authors did not discuss their contributions in detail.
(2) The proposed algorithm still can be improved if the ideas in the following papers are explored, i.e., "rank-constrained spectral clustering with flexible embedding", "dynamic affinity graph construction for spectral clustering using multiple features", and "zero-shot event detection via event-adaptive concept relevance mining". The authors are encouraged to discuss them in the revision.
(3) The authors should carefully proofread this paper and correct all the typos in the revision. In the current version, there are still some typos/grammar errors.
(4) Could the authors report the running time of the proposed algorithm? In this way, we can justify whether this algorithm can be applied to large-scale dataset.
Based on the above comments, I would like to accept this paper with major revision.
Author Response
Kindfull see the attachment

Reviewer 2 Report
This paper presents an interesting approach for understanding the quality and ripeness of agricultural products.
The method is based on DPP and is applied to 'Galia’ muskmelons. The proposed testing method is the use of acoustic resonance testing (ART).
The paper is very interesting and nicely written.
Some improvement ideas:
- the reader might expect to find in 2.2. Experimental Design the methodology steps for the research. Otherwise the title could be changed. Yet, here would be the place where to describe all these steps in some further details.
- in the final chapter, 4. Discussion, some links towards attaining the research goals, should be made
- "[42] showed that the configuration of the architecture can strongly influences the quality of the acquisition function"
\> should be rephrased so not to start with a reference
Author Response
Kindly see the attachment

Round 2
Reviewer 1 Report
The authors have carefully addressed all of my concerns. I have no further concerns now, and recommend to accept it as it is.